# Research on Development 3D Ground Penetrating Radar Acquisition and Control Technology for Road Underground Diseases with Dual-Band Antenna Arrays

**DOI:** 10.3390/s23198301

**Published:** 2023-10-07

**Authors:** Liang Fang, Feng Yang, Maoxuan Xu, Fengyu Liu

**Affiliations:** School of Mechanical Electronic and Information Engineering, China University of Mining and Technology, Beijing 100083, China; yangf@cumtb.edu.cn (F.Y.); xmx@student.cumtb.edu.cn (M.X.); bqt2100405024@student.cumtb.edu.cn (F.L.)

**Keywords:** road underground disease detection, 3D GPR, time division step multiplexing, dual-band antenna arrays

## Abstract

This paper describes the development of a new 3D ground-penetrating radar (GPR) acquisition and control technology for road underground diseases with dual-band antenna arrays. The 3D GPR system can be mounted on a vehicle-loading device and used by vehicles to detect road underground diseases at regular speeds. Compared with existing 3D GPR systems, this new type of 3D GPR has the following design features: it has dual-band antenna arrays, including a 16-channel 400 MHz antenna array and an 8-channel 200 MHz antenna array, which not only improves the detection efficiency, but also effectively balances the detection depth and detection resolution. A novel antenna switching method for time division step multiplexing (TDSM) is realized via field programmable gate array (FPGA), which not only avoids the crosstalk of antenna echo signals of different frequencies, but also ensures the interval of the same antenna working time. By combining the advantages of the FPGA and micro-control unit (MCU), and utilizing the high-speed transmission of the network port, the high-speed real-time transmission of the 3D GPR echo data is achieved. Finally, the integration of all software and hardware verified the correctness of the system, with good results.

## 1. Introduction

GPR, as a recognized non-destructive testing tool, has been widely used in the detection of underground road diseases, effectively detecting diseases such as settlement, looseness, water-rich areas, cavities, etc. Among them, the 3D GPR technology has obvious advantages: it can collect high-density and uniformly spliced massive radar data, reducing the lack of underground information; true 3D acquisition; it can make the imaging of underground targets clear and accurate; and its three-dimensional data volume can achieve horizontal slice display at any depth [1,2].

At present, the main control method for 3D ground-penetrating radar is time division step multiplexing (TDM), and only a single-frequency antenna array is used. The TDM method cannot avoid the continuous reception of echo signals by the same receiving antenna, resulting in the receiving antenna working continuously and waiting for too long. These systems detect highway underground diseases with different depths and resolutions, and need to replace antenna arrays with different frequencies for multiple measurements. Moreover, it is challenging to compare and process the 3D data of multiple measurements of antenna arrays with different frequencies, which is inefficient for road damage detection. For example, MIRA, a 3D GPR array in Mala, Sweden, used the TDM method, which provided a standard system of 400 MHz antenna array, even with 200 MHz or 1.3 GHz antenna arrays available for selection [3]. The Stream series 3D GPR array developed by IDS in Italy also used the TDM method, and both the stream X system and stream C system were only equipped with a 200 MHz single band antenna array or a 600MHz single band antenna array [4,5,6]. The GeoScope 3D GPR system developed by the Norwegian 3D-Radar company, and included an antenna array that can step at frequencies ranging from 200 MHz to 3 GHz, as well as collect and store data in a frequency domain [7,8,9]. Theoretically, the data in the frequency domain have many central frequencies and contain more information, but this is only a theory. The ZRY-TD5200 of Dalian Zoroy uses the TDM method as well, and only provides a 400 MHz dual-polarized antenna array with a common offset [10].

To address the limitations of the aforementioned 3D GPR, a new 3D GPR acquisition and control technology for detecting underground road diseases with dual-band antenna arrays has been designed and developed, which includes 200 MHz and 400 MHz dual-band antenna arrays. In order to avoid signal interference between different frequencies and to use the same receiving antenna for short multiplexing time intervals, a novel switch switching method of TDSM was implemented using FPGA precise control. By combining the advantages of precise parallel control with FPGA, as well as the efficient integration of A/D converters and data caching within the MCU, equipped with gigabit network communication, a local area network is established to achieve a high-speed real-time transmission of 3D GPR echo data. The high- and low-frequency characteristics can ensure high resolution in shallow layers and achieve deeper detection depths.

## 2. System Architecture

This 3D GPR mainly includes the following parts: dual-band antenna arrays, a control center, a data acquisition center, and data acquisition software. As shown in Figure 1.

The dual-band antenna arrays consists of 200 MHz and 400 MHz transceiver antenna groups, which adopts a dual row TE polarization common offset layout. The control center includes an FPGA core board and a new main control board. The FPGA core board mainly completes network communication logic, TDSM switch switching logic, DA control logic and digital output, data acquisition control and enable timing logic, and distance measurement wheel control logic. The new main control board is designed and configured with a switch switching circuit, DA conversion circuit, level comparison circuit and ranging wheel input circuit, which mainly complete the generation of delay step, the generation and output of transceiver pulse pair, and the pulse input of the ranging wheel. The data acquisition module of the MCU is equipped with a gigabit network port communication circuit, an A/D input circuit, and a control and enable time sequence input circuit, which mainly completes the data acquisition, buffering, and transmission of the echo signals of each channel. The data acquisition software is carried on the computer to realize and control the acquisition parameter configuration and data acquisition process of different channels through the local area network, and realize the real-time display of the horizontal section, vertical section, and longitudinal 2D section of the 3D GPR, as well as the data storage of each channel.

## 3. Dual-Band Antenna Arrays

### 3.1. Antennas Selection

The detection depth of GPR is related to the transmission power of the antenna, the frequency used, the conductivity characteristics of the medium, and the dynamic range of the instrument [11]. The conductivity characteristics of a medium are related to the characteristics of the medium itself. The indicator for reflecting the dynamic range is the sampling number, which is the conversion number of A/D, with 24 bit being the best, followed by 16 bit, and the worst being 8 bit.

The maximum depth at which a ground-penetrating radar can detect a target is called the detection depth of the GPR, also known as the detection distance [12,13]. When the radar system is selected, if the scattering cross-section of the target scatterer is ${\;\sigma }_{s}$, the gain of the scatterer toward the back is $\;G_{s}$, the directional gain of the receiving antenna is $G_{r}$, the efficiency of the transmitting and receiving antennas is, respectively, $\eta_{r}$ and $\eta_{t}$, the antenna transmission power is $W_{t}$, the directional gain of the transmitting antenna is $G_{t}$, the wavelet length of the radar is λ, the electromagnetic wave propagation distance is r, and the absorption coefficient of the medium is b, then the power received by the radar antenna is
(1)$$W_{r}=\sigma_{s}G_{s}G_{r}\eta_{r}W_{t}\eta_{t}G_{t}\frac{\lambda^{2}}{4\pi^{3}r^{4}}e^{-4br}$$

From the above formula, it can be seen that the longer the sub wave length, the further the detection distance, and the wavelength is inversely proportional to the antenna frequency. Therefore, the further the detection distance, the lower the antenna frequency. GPR resolution is divided into horizontal resolution and vertical resolution [14,15].

Horizontal resolution is the minimum horizontal scale of an object that a GPR can distinguish. Horizontal resolution is a crucial technical indicator for engineering exploration. According to the Fresnel principle, the difference in wave path between the center vertical reflection and the edge reflection of the Fresnel zone is *λ*/2. Assuming that the mine detection wave propagates downward in the form of a cone, and most of the energy is reflected back from the surface of the object, the horizontal resolution can be estimated according to the following equation:(2)$$r=\sqrt{\left(h\lambda +\frac{\lambda^{2}}{4}\right)}$$
where r is the radius of the cylinder, λ is the electromagnetic wave length, and h is the column top buried depth. The horizontal resolution should be 1/2 of the Fresnel zone radius *r* of the cylinder radius.

Vertical resolution is the minimum vertical scale of an object that can be detected. According to the interference theory of waves, the minimum recognizable bidirectional wave path difference of the reflected waves from the upper and lower interfaces of an object is λ/8~λ/4; thus, the vertical resolution $\;R_{v}$ is related to antenna wavelength: $R_{v}=\lambda /8$~λ/4. It can be seen that both horizontal and vertical resolutions are related to antenna wavelength, while wavelength is determined by antenna frequency and medium wave velocity.

Based on application practice and detection distance formula, the following table provides the range of detection depths and resolutions at different frequencies for highways (assuming a dielectric constant of 9).

According to Table 1, in order to achieve the detection targets of highway roadbed structure and foundation, as well as an ultra 3 m underground damage detection, and with good resolution, it is appropriate to select a combination of 200 MHz and 400 MHz antennas with a center frequency. This study used shielded bowtie antennas designed by the Urban Underground Space Safety Research Center of China University of Mining and Technology (Beijing) with central frequencies of 400 MHz and 200 MHz.

### 3.2. Antenna Layout

According to the width of the motor vehicle lane, it should not be less than 3 m according to the provisions of Technical Standards for Highway Engineering (JTG B01), so the width of on-board antenna array should not exceed 3 m. The size of the selected 400 MHz antenna is 280 mm in length, 120 mm in width, and 90 mm in height, as show in Figure 2a. The 200 MHz antenna is 420 mm in length, 350 mm in width, and 200 mm in height, as show in Figure 2b. The shielding box structure is shown in Figure 2c. Therefore, when using TE polarization direction, the 400 MHz antenna layout cannot exceed 3 m/280 mm ≈ 10, and the 200 MHz antenna layout cannot exceed 3 m/420 mm ≈ 7.

The results of the return loss S11 measured using a digital serial analyzer are shown in Figure 3. It can be clearly observed that the return loss of the bowtie antenna is less than −10 dB.

Based on the above antenna, due to the influence of the shielding shell and shielding material, this 3D ground-penetrating radar has designed an antenna layout as shown in Figure 4. Nine transmitting antennas and eight receiving antennas with a center frequency of 400 MHz were selected, and four transmitting antennas and five receiving antennas with a center frequency of 200 MHz were selected, totaling 13 transmitting antennas and 13 receiving antennas. Double-row common offset is adopted, and the ground coupling TE polarization direction is arranged [16,17]. This double-row layout not only avoids the electromagnetic interference between antennas with different frequencies, but also ensures that different antenna pairs with the same frequency are equidistant, which makes the later data processing simple.

## 4. Control Center

To achieve the operation of a dual-band antenna array, the control center of this 3D GPR is designed, consisting of a combination of an FPGA core board and a new main control board. As shown in Figure 5a, the FPGA core board uses the black gold AX415-Cyclone IV core development board, with a EP4CE15E17C8N chip, which has low power consumption and strong anti-interference [18,19]. It mainly completes network communication logic, TDSM switch switching logic, DA control logic, step-delay digital output, acquisition control and enable timing logic, and distance measurement wheel control logic. Based on the dual-band antenna array, a new type of main control board has been designed and developed in this study, as shown in Figure 5b, including DA conversion circuit, level comparison circuit, switch switching circuit, ranging wheel input circuit, and power circuit.

### 4.1. Control Center Workflow

The workflow of the control center is shown in Figure 6. Firstly, establish network communication and connect to the data acquisition software. Then, according to the parameters set by the data acquisition software, use registers and latches to store the step delay and other parameters of each channel, and generate a work cycle based on the set sampling frequency, sampling points, and other parameters. During each working cycle, the working channel is opened according to the switch switching method, and then the FPGA completes the step-delay control logic for that channel. That is, based on the step-delay parameters stored in the register, the calculated step-delay digital quantity is sent to the DA converter to generate a step-delay voltage. Then, through the level comparison circuit, the transmitting and receiving pulse pairs of this channel are generated and sent to the corresponding transmitting and receiving antenna pairs in the dual-band antenna array. Then, a data acquisition enable is generated and sent to the corresponding data acquisition center MCU of this channel, completing the acquisition logic of each point in one channel, switching to the next channel, and repeating the above process until all channels have completed acquisition. Repeat the above process based on the number of sampling points until all channel sampling points have been collected, and the data of the set sampling points are obtained. Finally, send the data to the data collection center to enable sending.

### 4.2. TDSM Switch Switching Method

Sequential equivalent sampling is to obtain and reconstruct the signal waveform by triggering multiple sampling times. The premise of equivalent sampling for a signal is that it is a periodic signal [20,21]. The pulse frequency of GPR is high, so it is assumed that the electromagnetic wave emitted by a limited number of pulse antennas is almost periodic information; that is, every time the signal is transmitted, it can be regarded as a repeated signal according to the pulse frequency because the surrounding environment has not changed. Therefore, GPR meets the premise of equivalent sampling.

The echo signals of different distances and frequencies of the transmitting and receiving antennas at the same time will inevitably interfere with each other, so the general antenna array uses time division multiplexing (TDM) to use each transmitting and receiving antenna pair; that is, only one antenna pair is turned on at a time, and then the next antenna pair is turned on after sampling is completed [22,23,24].

This 3D GPR adopts 24 channels, and according to the sequential equivalent sampling, if the sampling frequency is 200 KHz, the sampling period of all channels will be 1/200 KHz = 5 us; that is, all logic control and data reading and output of 24 channels will be completed within 5 us. So if using TDM sequence to enable each group of antenna pairs, the working cycle of a single channel is 5 us/24 = 0.2083 us, and the actual working frequency or sampling rate is 4.8 MHz, which is much higher than the highest operating frequency of 3.6 MHz of the built-in A/D converter in MCU, which is obviously not suitable. The antenna array adopts the common offset layout; that is, the distance between the receiver and the transmitter at the same frequency is certain. In order to reduce the sampling rate, it can be regarded as the mode of one transmitter and two receivers, so the actual working time of a single channel is only related to the number of transmitters; that is, a single channel only needs to complete all logic control and data reading and output within the working cycle of 5 us/13 = 0.3846 us.

If the TDM switch switching method is used, the sequential switch turns on 13 transmitters, that is, T1 to T13 sequential switches are switched and recycled, as shown in Table 2. It can be seen from the table that two consecutive actual working hours of the same receiver will spare another 11 actual working hours, which is a burden and waste to the receiver.

To address the issue of short time intervals between the receiving antennas of the switches mentioned above, an improved TDM switch switching method was proposed, and a TDSM switch switching method was designed. That is, 13 transmitters are turned on by using the step-by-step switch mode; that is, after the T1 antenna group is switched on, it is switched to T4 step by step, and then to T7, etc., step by step. After switching all antenna groups, it is recycled to T1, as shown in Table 3. Each antenna pair is switched step by step, and it is circularly time division multiplexing. It can be seen that the shortest working time interval of the same receiver is 0.3846 us × 3 = 1.1538 us.

As shown in Figure 7, the switching timing of TDSM switch is realized by FPGA, with high voltage indicating that the switch is on, low voltage indicating that the switch is off, ct[n−1] corresponds to Tn, and cr[n−1] corresponds to Rn, so it can be seen that the switching time interval between the receiver and the transmitter is even.

To verify the TDSM effect of the control center switch switching logic control module, it is necessary to use an oscilloscope to observe the timing relationship of the output clocks of each channel. Considering the symmetry of the module and the limitation of the number of oscilloscope channels, the transmitting switches T7 and T8, and the receiving switches R7 and R8, were selected to be combined and connected to the oscilloscope (TBS1102B EDU) for observation. The waveform shown in Figure 7 can be obtained. Figure 8a shows the opening time of a single switch of 0.384 us, which is consistent with theoretical calculations. Figure 8b shows the opening interval of 1.92 us for the front and rear transmission antenna switches (selecting T7 and T8). Figure 8c–e shows a comparative display of the opening sequence of a transmitting antenna switch (T8) and its corresponding two receiving antenna switches (R7 and R8), indicating that the opening interval of the receiving antenna is consistent with the design requirements, with a minimum of 1.153 us. Furthermore, it can be observed that the cycle of each switch sequence is 5us, achieving a design switching frequency of 200 KHz.

### 4.3. Step Delay

For GPR systems using equivalent sampling methods, a high precision step delay circuit is extremely important, as it determines the accuracy of the ground penetrating radar signal acquisition system [25,26]. This 3D GPR adopts a shared DA conversion circuit, where all the transmitting and receiving pulse pairs are compared with the DA conversion circuit of the previous stage of the circuit. By using a DA chip, 24 pulse pairs are generated, reducing costs. In Figure 9, DA represents a 16 bit DA converter circuit that provides fast ramp switching voltage and delay voltage. After the DA converter generates a step delay voltage, it passes through a level comparison circuit and a time-sharing step multiplexing switch to generate a pulse pair sent to the receiving and transmitting antenna, as shown in Figure 10. Figure 10a–c show a single channel receiving and transmitting a pulse, and Figure 10d–f show a transmitting pulse and its corresponding two receiving pulses.

## 5. Data Acquisition Center

As shown in Figure 11, four groups of MCU are combined to form a data acquisition center. In this study, a new data acquisition board is developed, and the MCU chip is STM32AH743. As shown in Figure 12, the acquisition board includes a network communication circuit, A/D input circuit, acquisition control and enabling timing input circuit (which mainly completes the echo data acquisition), and caching and transmission of each channel. Three A/D converters are integrated in the selected MCU, and its maximum frequency is 3.6 MHz when the sampling bit is 16 bit. This 3D GPR has 24 channels. In order to make full use of the internal resources of a single-chip microcomputer, each A/D converter is selected to receive two channels of data, so each MCU collects six channels of data. The data of each channel are cached by using the stack data structure. After obtaining the data transmission control timing of FPGA, the six-channel data are obtained from the cache and packaged with the data header, sent to the switch, and then received by the data acquisition software. Each MCU sends the data of each channel in sequence, but the data sent by the four single chips are sent to the switch in parallel, without interference.

As shown in Figure 13, the MCU collects data according to the rising edge of the data sampling enabling sequence of each channel sent by the FPGA. One rising edge collects one data point, which is stored in the data cache data structure of each channel in sequence. After receiving the data transmission control sequence sent by the FPGA, it will take out the data in the cache, add the data header, pack it, and send it to the switch, and the data acquisition software will receive it. Then read the next set of data acquisition enable timing and data transmission enable timing for each channel until the FPGA stops send enable timing. Each MCU sends data from each channel in sequence, but the four sets of MCU modules send data in parallel to the local area network without interfering with each other.

## 6. Integration and Validation

According to the system structure of Figure 1, all the unit modules included in this 3D GPR are connected and arranged as shown in Figure 14. Among them, the control center, data acquisition center, and multi-frequency antenna array are assembled and installed in the loading device as shown in Figure 15 at the back of the vehicle, which is connected to the switch in the vehicle by a network cable, and the computer loaded with data acquisition software is installed in the vehicle, which is also connected to the switch by a network cable, thus realizing the integration of the 3D GPR system. In the process of detection, the operator drives the vehicle to scan the target area and obtain the required GPR data. When the system is triggered, a complete set of 24-channel data, including their position, can be collected every 3 milliseconds at the fastest. Under the requirement of ensuring imaging accuracy, the vehicle speed can reach 70 km/h, allowing the vehicle to measure its normal speed on the road. The comparison table of the working parameters of the dual-band antenna array included in the 3D ground-penetrating radar vehicle is shown in Table 4.

The data acquisition software includes communication and acquisition control, aiming to achieve and control the configuration of acquisition parameters and data acquisition process for different channels. The collection control and data display interface is shown in Figure 16a. The acquisition control buttons in the figure include antenna selection and acquisition process selection buttons. Antennas can be selected to have all antennas turned on, with only 400 MHz antennas turned on or only 200 MHz antennas turned on, while the rest are acquisition control selection buttons. Channel selection is used to select the grayscale image of the displayed channel in the binary profile. The horizontal grayscale image is displayed in real-time on a horizontal slicing screen. The vertical grayscale image is displayed in real-time on a vertical slicing screen. The grayscale image of the selected channel is displayed in real-time on a 2D profile screen. The horizontal and vertical tangents represent the real-time positional tangents of the horizontal and vertical profiles. The horizontal tangent can be moved up and down with the mouse button to select the depth of the horizontal section. The interface for collecting parameters is shown in Figure 16b. Through this interface, set the collection parameters of all channels and transfer the parameters to an FPGA based on network ports and switches to prepare for data collection.

Through the integration of various modules of the new 3D GPR system mentioned above, the acquisition of echo data from various channels has been achieved. The data acquisition software has also completed the real-time display of horizontal, vertical, and vertical two-dimensional profiles, verifying the feasibility of the system. The new 3D GPR system is specifically designed for detecting underground road diseases. To demonstrate its performance, the key parameters of this system are compared with other 3D GPR systems used for detecting underground road diseases, as shown in Table 5. From Table 5, it can be seen that this system has two types of central-frequency antennas that can work simultaneously for detection, greatly improving investigation efficiency.

## 7. Conclusions

This article designs and develops a new 3D GPR acquisition and control technology for detecting underground road diseases with dual-band antenna arrays, which are installed and integrated into the vehicle mounted device. It includes a 200 MHz and 400 MHz dual-band antenna array, and its high- and low-frequency characteristics can ensure high resolution in shallow layers and achieve a deeper detection depth. In order to avoid signal interference between different frequencies and avoid an excessively short switching time for the same receiving antenna, a new switching method of TDSM was implemented using FPGA precise control. By combining the advantages of precise parallel control with FPGA, as well as the efficient integration of A/D converters and data caching within the MCU, and equipped with gigabit network communication, a local area network is established to achieve a high-speed real-time transmission of 3D GPR echo data. The real-time display of the data collection software verifies the integrity of the design. Next, we will test the stability and accuracy of the system, as well as the return loss of the dual-band ground-penetrating radar, and improve the data processing software to integrate various collected data for easy interpretation and damage feature extraction [27,28].

## Figures and Tables

**Figure 1 sensors-23-08301-f001:**
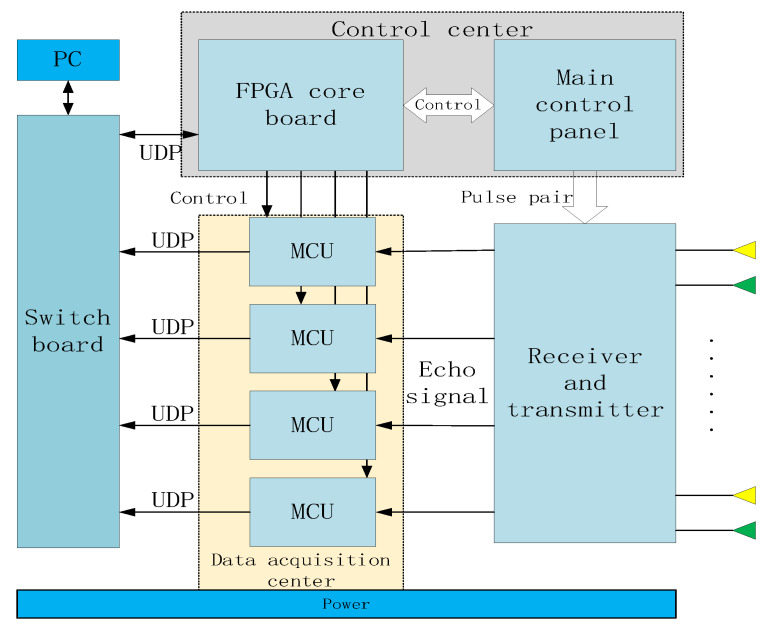
Architecture of dual-band antenna 3D GPR system.

**Figure 2 sensors-23-08301-f002:**
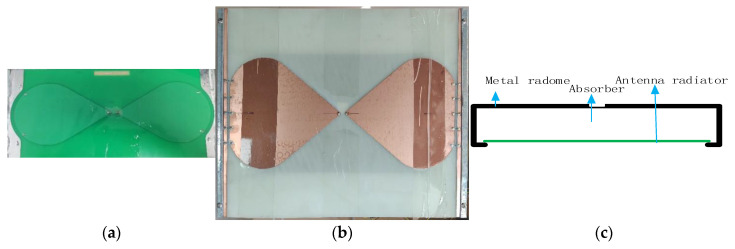
Bowtie antenna: (**a**) 400 MHz; (**b**) 200 MHz; (**c**) metal shielding box structure.

**Figure 3 sensors-23-08301-f003:**
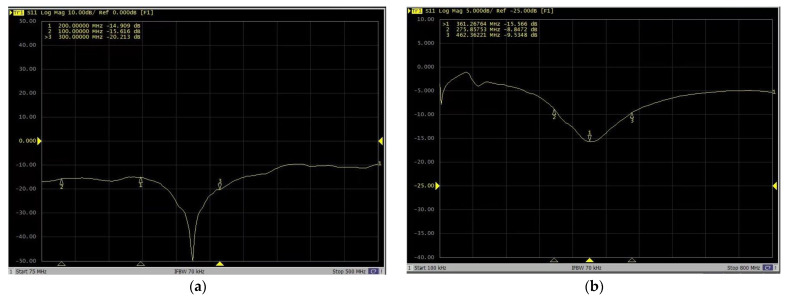
Bowtie antenna echo loss diagram: (**a**) 200 MHz; (**b**) 400 MHz.

**Figure 4 sensors-23-08301-f004:**
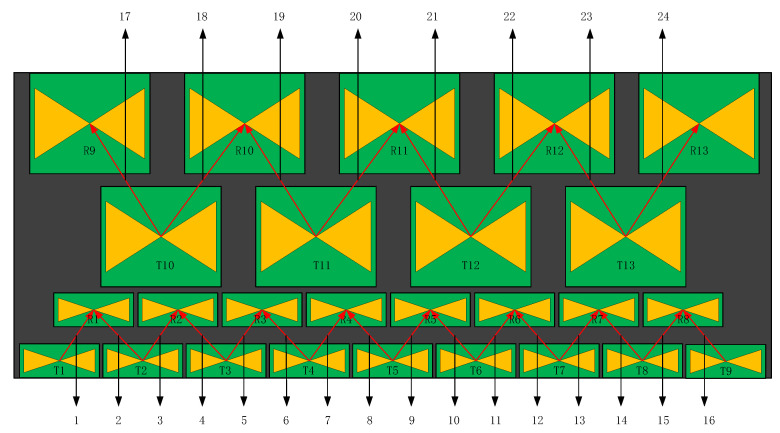
Layout diagram of complex-frequency antenna array.

**Figure 5 sensors-23-08301-f005:**
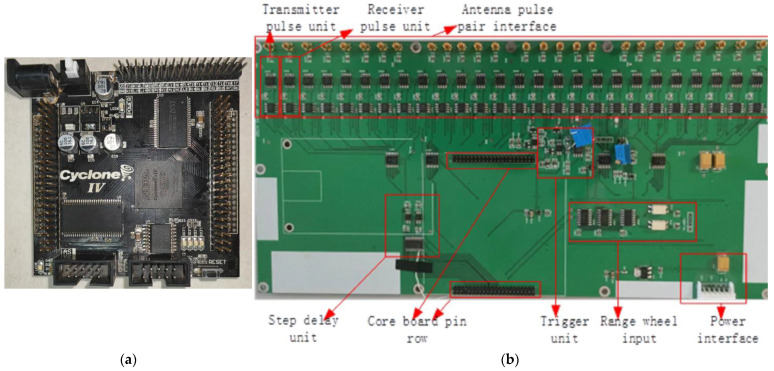
Control center board. (**a**) FPGA core development board; (**b**) New main control board.

**Figure 6 sensors-23-08301-f006:**
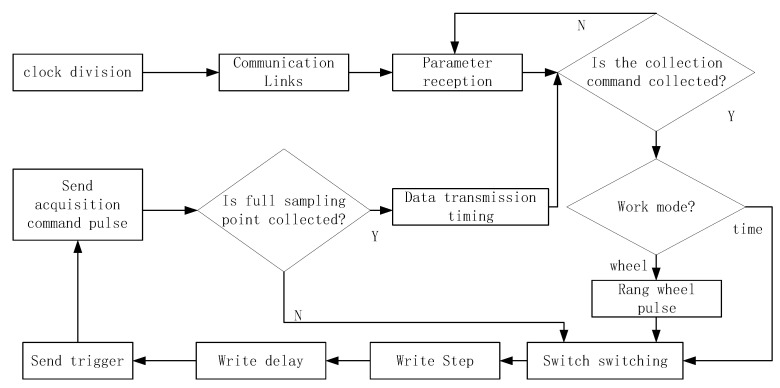
FPGA control logic flowchart.

**Figure 7 sensors-23-08301-f007:**
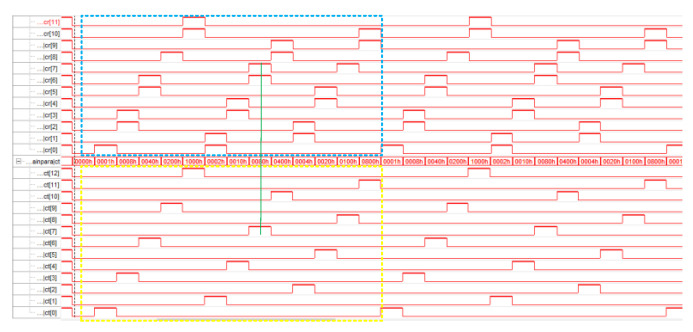
Implementation of TDSM switch switching FPGA timing.

**Figure 8 sensors-23-08301-f008:**
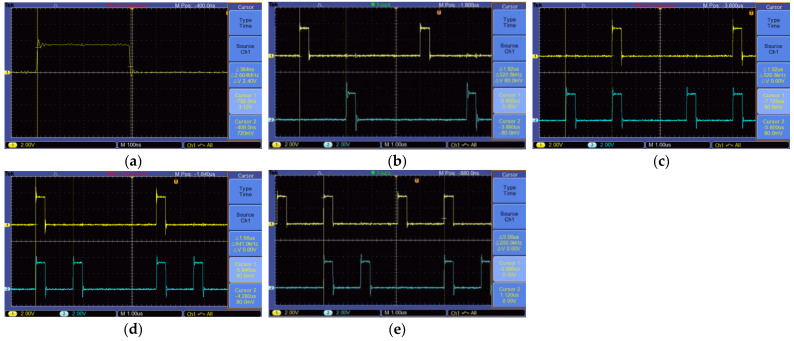
Switch timing test results. (**a**) The switch on time is 384 ns. (**b**) Channel 1: Transmit antenna T7 switch timing; Channel 2: Transmit antenna T8 switch timing. (**c**) Channel 1: Transmit antenna T8 switch timing; Channel 2: Receive antenna R7 switch timing. (**d**) Channel 1: Transmit antenna T8 switch timing; Channel 2: Receive antenna R8 switch timing. (**e**) Channel 1: Receive antenna R7 switch timing; Channel 2: Receive antenna R8 switch timing.

**Figure 9 sensors-23-08301-f009:**
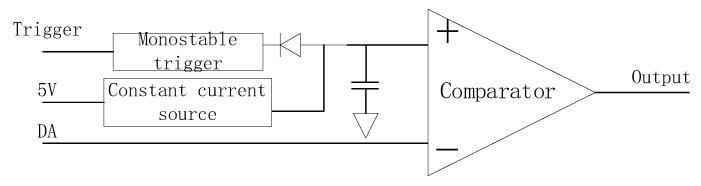
Step-delay structure diagram.

**Figure 10 sensors-23-08301-f010:**
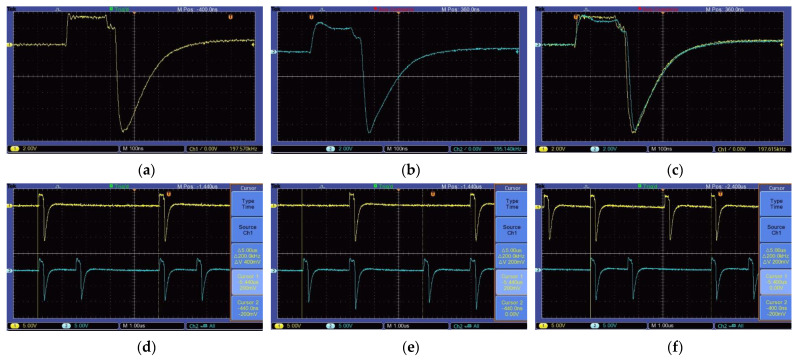
Transceiver pulse pairs: (**a**) Transmit antenna pulse; (**b**) Receive antenna pulse; (**c**) Channel 1: Transmit antenna pulse; Channel 2: Receive antenna pulse; (**d**) Channel 1: Transmit antenna T8 pulse; Channel 2: Receive antenna R7 pulse; (**e**) Channel 1: Transmit antenna T8 pulse; Channel 2: Receive antenna R8 pulse; (**f**) Channel 1: Receive antenna R7 pulse; Channel 2: Receive antenna R8 pulse.

**Figure 11 sensors-23-08301-f011:**
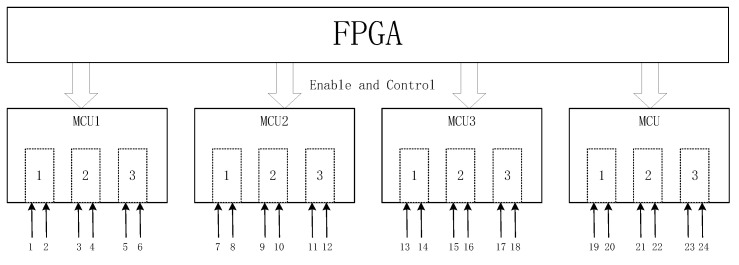
Data collection center.

**Figure 12 sensors-23-08301-f012:**
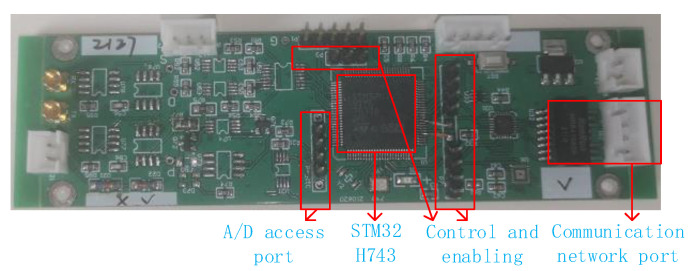
MCU module.

**Figure 13 sensors-23-08301-f013:**
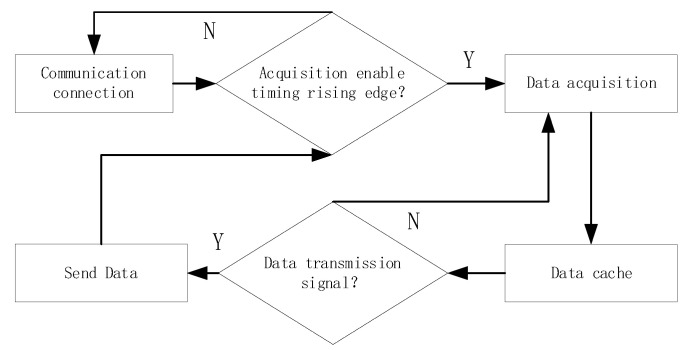
Data collection flowchart.

**Figure 14 sensors-23-08301-f014:**
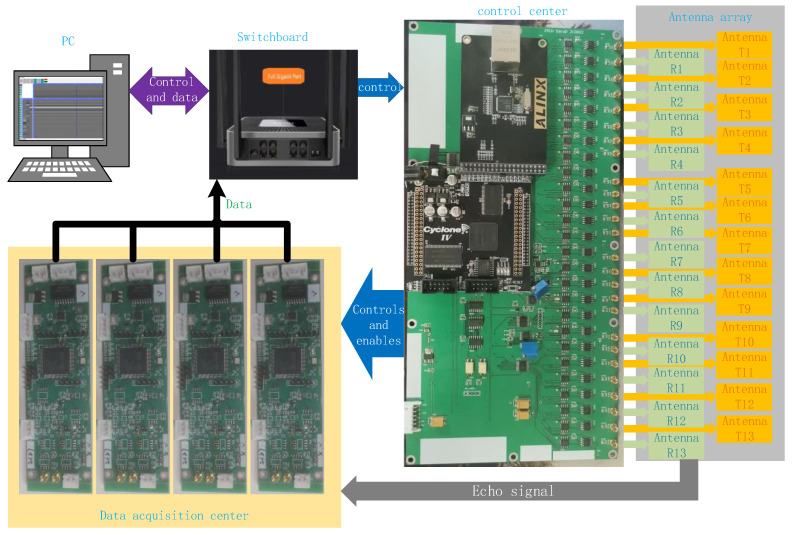
Integrated layout of 3D ground-penetrating radar.

**Figure 15 sensors-23-08301-f015:**
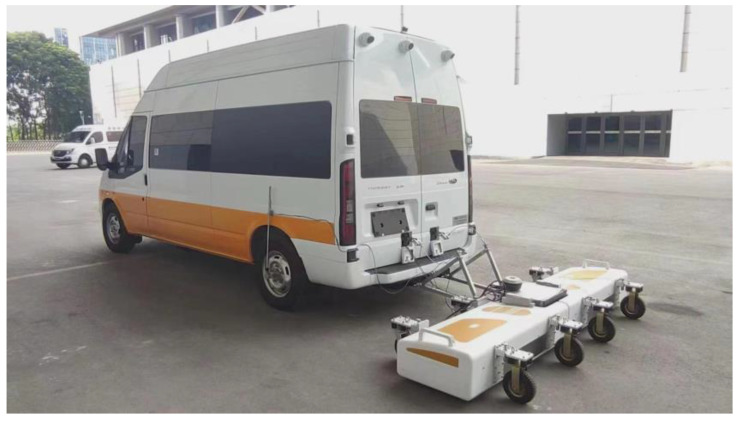
Three-dimensional ground-penetrating radar vehicle diagram.

**Figure 16 sensors-23-08301-f016:**
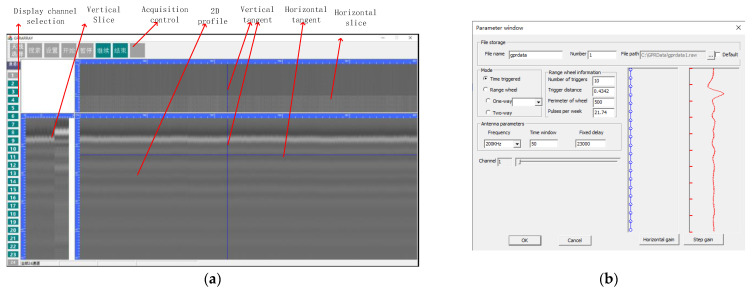
Data acquisition software interface. (**a**) Control and display interface. (**b**) Parameter setting interface.

**Table 1 sensors-23-08301-t001:** Detection depth and resolution range of antennas with different center frequencies.

Antenna	Investigation Depth (m)	Detect Target	Detect Target Size (Horizontal Resolution) (m)	Detect Target Size (Vertical Resolution) (m)
1600 MHz	0.6~0.8	Surface thickness and surface fall through	0.13	0.008~0.016
900 MHz	1.0~1.2	Roadbed structure	0.20	0.014~0.028
400 MHz	1.0~3.0	Roadbed structure and foundation	0.45	0.031~0.063
200 MHz	1.0~5.0	Roadbed structure and foundation	0.81	0.063~0.125
100 MHz	1.0~8.0	Roadbed structure and foundation	1.44	0.125~0.250

**Table 2 sensors-23-08301-t002:** TDM enables antenna pairs.

Number	1	2	3	4	5	6	7	8	9	10	11	12	13
Transmitting antenna switch	T1	T2	T3	T4	T5	T6	T7	T8	T9	T10	T11	T12	T13
Receiving antenna switch	R1	R1	R2	R3	R4	R5	R6	R7	R8	R9	R10	R11	R12
		R2	R3	R4	R5	R6	R7	R8		R10	R11	R12	R13

**Table 3 sensors-23-08301-t003:** TDSM enables antenna pairing.

Number	1	2	3	4	5	6	7	8	9	10	11	12	13
Transmitting antenna switch	T1	T4	T7	T10	T13	T2	T5	T8	T11	T3	T6	T9	T12
Receiving antenna switch	R1	R3	R6	R9	R12	R1	R4	R7	R10	R2	R5	R8	R11
		R4	R7	R10	R13	R2	R5	R8	R11	R3	R6		R12

**Table 4 sensors-23-08301-t004:** Comparison table of working parameters.

Antenna Type	Antenna Array Main Frequency	Pulse Width	Reachable Depth	Reference Depth	Application
Single shielded antenna	200 MHz	5 ns	1–5 m	3 m	Roadbed structure and foundation
	400 MHz	2.5 ns	1–3 m	2 m	Roadbed structure and foundation

**Table 5 sensors-23-08301-t005:** Comparison of key parameters of 3D ground-penetrating radar.

Type	Novel 3D GPR System	Stream X
Number of channels	24	16
Pulse repetition frequency	200 KHz	50 KHz
Center frequency of antenna array	200 MHz and 400 MHz	200 MHz
Maximum sample rate	7992 scans/s while 512 samples per scan	1450 scans/s while 512 samples per scan
Antenna array layout	Common offset	Dipoles parallel to the forward (vertical polarization) direction
Antenna switching method	TDSM	TDM

## Data Availability

Not applicable.

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
