# Peer review of "Research on Development 3D Ground Penetrating Radar Acquisition and Control Technology for Road Underground Diseases with Dual-Band Antenna Arrays"

_sensors, 2023, doi:10.3390/s23198301_

Round 1
Reviewer 1 Report
In this paper, a ground-penetrating radar controller for a dual-antenna array is designed to realize antenna switching using the TDSM method. In order to better reflect the innovation and workload of the article, the following suggestions are requested to be stated by the authors in the paper
1, The effect of the spacing of neighboring transmitter antennas of the radar array on the switching time?
2, How is the spacing of neighboring antennas of the radar array determined, and is there any duplicate scanning?
3, What is the detection depth of the radar array?
4, System stability and accuracy test?
5, What is the time utilization of the TDSM switching method in this paper?
6, How does the control switching method compare with other existing multiband radars?
7, How does the radar system in this paper compare with other 3D GPR Systems in terms of test performance?
Author Response
Thank you very much for taking the time to review this manuscript. As you are concerned, there are several problems that need to be addressed. According to your nice suggestions, we have made extensive corrections to our previous draft, the detailed responses to you are listed below.
Comments 1: The effect of the spacing of neighboring transmitter antennas of the radar array on the switching time?
Response 1: Thank you for your suggestion. The TDSM method we designed is an improvement on the foundation of the TDM method. The TDM method separates the crosstalk signal between transmitting antennas and the effective radar echo in time. Only one transmitting antenna is opened at a time, and the next antenna is opened after sampling is completed. Therefore, at a fixed sampling frequency, the switching time is only affected by the number of transmitting antennas, and is not affected by the spacing between transmitting antennas in space.
Comments 2: How is the spacing of neighboring antennas of the radar array determined, and is there any duplicate scanning?
Response 2: According to section 3.2 (page 4 of the manuscript), we have simply determined the spatial position of the antenna based on the selected dual band antenna structure, lane width limitations, dual row co offset method, and grounding coupling TE polarization direction layout method. We have not conducted any experiments on the influence of adjacent antenna spacing. Thank you for your valuable advice. Due to time constraints, we were unable to add it to the manuscript in a timely manner. We hope for your forgiveness. The radar has not detected any duplicate scanning issues during use.
Comments 3: What is the detection depth of the radar array?
Response 3: Thank you for your question. We mentioned the detection depth of 200MHz and 400MHz antennas in Table 1 (page 4 of the manuscript), and the detection depth of radar arrays can refer to this value.
Comments 4: System stability and accuracy test?
Response 4: Thank you for your suggestion and reminder. System stability and accuracy testing are indeed what we lack and what we need to do next. We attach the system stability and accuracy testing as an outlook at the end of the conclusion section.
Comments 5: What is the time utilization of the TDSM switching method in this paper?
Response 5: After reviewing relevant materials and literature, we were unable to fully understand what you said about the time utilization rate. For our design of TDSM switching method, at a sampling frequency of 200KHz, the total opening time of all channels is 5us, and the opening time of each switch is 0.384us. The same is true for the oscilloscope measurement results in Figure 7 (a) (page 8of the manuscript).
Comments 6: How does the control switching method compare with other existing multiband radars?
Response 6: Thank you for your suggestion. The TDSM switching method is an improvement of the TDM method, which separates channel to channel crosstalk signals and effective radar echoes in terms of time. The improvement also avoids the defect of the same receiving antenna being turned on for too short a time interval. Other existing 3D ground penetrating radars are mainly single band, using TDM method. Added the switching method column in Table 4(page 12of the manuscript ) .
Comments 7: How does the radar system in this paper compare with other 3D GPR Systems in terms of test performance?
Response 7: Thank you for your request and suggestion. We have selected the Stream X 3D ground penetrating radar mentioned in the introduction and made a partial comparison with our 3D ground penetrating radar. The specific content can be found in Table 4 (page 12 of the manuscript).
We tried our best to improve the manuscript and made some changes marked up using the “Track Changes” function in revised manuscript. After revising, the quality of this manuscript is highly improved. And the expression of the article is clearer than the original manuscript. If you think there is any need for further modification, we will actively cooperate. We hope you will find our revised manuscript acceptable for publication.
Reviewer 2 Report
1. Index term should be in alphabetical order.
2. “The interface for acquisition control and data display is shown in Figure 15(a).” Author should check Fig.15 (a) and (b). Similarly Fig.14.
3. The author should add recent quality papers. Show the comparison of proposed work with recent and closely related papers.
Author Response
Thank you very much for taking the time to review this manuscript. As you are concerned, there are several problems that need to be addressed. According to your nice suggestions, we have made extensive corrections to our previous draft, the detailed responses to you are listed below.
Comments 1: Index term should be in alphabetical order.
Response 1: We apologize for our carelessness. We carefully checked and found that the index table of contents for the second paragraph on page 11 of the manuscript was not written in alphabetical order, and the paragraph has been revised.
Comments 2: “The interface for acquisition control and data display is shown in Figure 15(a).” Author should check Fig.15 (a) and (b). Similarly Fig.14.
Response 2: Thank you for your correction. Figures 14 (page 11), Figures 15 (a) (page 12), and Figures 15 (b) (page 12) in the manuscript are indeed not clear enough. We have optimized and replaced these three figures.
Comments 3: The author should add recent quality papers. Show the comparison of proposed work with recent and closely related papers.
Response 3: Thank you for your suggestion. We added two citations at the end of the literature citation to improve the comparability of the manuscript. Cited as “E. Balzovsky, Y. Buyanov, A. V. Yurchenko and V. I. Syryamkin, "Research and Development of the Antenna Array for Ground Penetrating Radar," Matec Web of Conferences, vol. 79, p. 1036, 2016.” and “X. Bai, Y. Yang, Z. Wen, S. Wei and J. Zhang et al., "3D-GPR-RM: A Method for Underground Pipeline Recognition Using 3-Dimensional GPR Images," Applied Sciences, vol. 13, no. 13, p. 7540, 2023.”
We tried our best to improve the manuscript and made some changes marked up using the “Track Changes” function in revised manuscript. After revising, the quality of this manuscript is highly improved. And the expression of the article is clearer than the original manuscript. If you think there is any need for further modification, we will actively cooperate. We hope you will find our revised manuscript acceptable for publication.
Reviewer 3 Report
The authors demonstrated research on Development of 3D Ground-Penetrating Radar with Dual-Band Antennas for Detecting Underground Road Diseases. I have some technical comments, which are as follows:
1- The introduction needs improvement. The authors should provide more information about the antennas being used for GPR applications. Additionally, please discuss the antenna parameters, such as bandwidth and gain [1,2]. Please refer to these references as they may add value to the introduction.
- Vivaldi Antenna Arrays Feed by Frequency-Independent Phase Shifter for High Directivity and Gain Used in Microwave Sensing and Communication Applications. Sensors 2021, 21, 6091. https://doi.org/10.3390/s21186091.
- Compact Size and High Gain of CPW-Fed UWB Strawberry Artistic Shaped Printed Monopole Antennas Using FSS Single Layer Reflector," in IEEE Access, vol. 8, pp. 92697-92707, 2020, doi: 10.1109/ACCESS.2020.2995069.
2- The authors mentioned the term 'dual-band antenna.' However, it's not clear from the context how we can determine that it is indeed a dual-band antenna. Additionally, could you please provide information on the frequencies at which the antenna operated?
3- The graph of antenna Return loss is mission?
4- Comparison table with related work is also missing.
That's all for me at this moment! The authors are required to revise the comments above carefully. Thanks
There are minor errors that need to be carefully checked!
Author Response
Thank you very much for taking the time to review this manuscript. As you are concerned, there are several problems that need to be addressed. According to your nice suggestions, we have made extensive corrections to our previous draft, the detailed responses to you are listed below.
Comments 1: The introduction needs improvement. The authors should provide more information about the antennas being used for GPR applications. Additionally, please discuss the antenna parameters, such as bandwidth and gain [1,2]. Please refer to these references as they may add value to the introduction.
Vivaldi Antenna Arrays Feed by Frequency-Independent Phase Shifter for High Directivity and Gain Used in Microwave Sensing and Communication Applications. Sensors 2021, 21, 6091. https://doi.org/10.3390/s21186091.
Compact Size and High Gain of CPW-Fed UWB Strawberry Artistic Shaped Printed Monopole Antennas Using FSS Single Layer Reflector," in IEEE Access, vol. 8, pp. 92697-92707, 2020, doi: 10.1109/ACCESS.2020.2995069.
Response 1: Thank you for your suggestion. We have carefully read the literature you provided, and our introduction to antennas is indeed insufficient. We apologize for any inconvenience caused. The dual band antenna array in our manuscript is composed of multiple combinations of 200MHz and 400MHz antennas. Our main focus is on the control of the dual band antenna array, which was studied by the upstream team using a single antenna. Therefore, we cannot present and discuss its antenna parameters in our manuscript. We have added the structure of a single butterfly antenna as shown in Figure 2 (fourth page of the manuscript) for better understanding.
Comments 2: The authors mentioned the term 'dual-band antenna.' However, it's not clear from the context how we can determine that it is indeed a dual-band antenna. Additionally, could you please provide information on the frequencies at which the antenna operated?
Response 2: Thank you very much for your question. The third section of our manuscript mainly introduces the reasons for selecting the two frequency antennas of the dual band antenna array and the layout of the dual band antenna array. In the following text, we did not display the information of the dual band antenna array very well. We have updated Figure 15 (a) to display a dual band antenna working sampling grayscale image in the vertical slice.
Comments 3: The graph of antenna Return loss is mission?
Response 3: Thank you very much for the questions and suggestions. We agree that more research or data will be helpful. We have indeed not done any work on antenna array return loss maps, as we cannot conduct this experiment in a short period of time. If the conditions are mature in the later stage, we will supplement the shortcomings in this area.
Comments 4: Comparison table with related work is also missing.
Response 4: Thank you for your request and suggestion. We have selected the Stream X 3D GPR mentioned in the introduction and made a partial comparison with our 3D GPR. The specific content can be found in Table 4 (page 12 of the manuscript).
We tried our best to improve the manuscript and made some changes marked up using the “Track Changes” function in revised manuscript. After revising, the quality of this manuscript is highly improved. And the expression of the article is clearer than the original manuscript. If you think there is any need for further modification, we will actively cooperate. We hope you will find our revised manuscript acceptable for publication.
Round 2
Reviewer 1 Report
Thanks to the authors for their responses and revisions, I have no further comments.
Author Response
Thank you for your suggestions on our manuscript, we have benefited greatly.
Reviewer 3 Report
The authors did not address any of my concerns. My comments were provided to enhance the quality of the paper, not for conversation!
1- The introduction needs improvement. The authors should provide more information about the antennas being used for GPR applications. Additionally, please discuss the antenna parameters, such as bandwidth and gain [1,2]. Please refer to these references as they may add value to the introduction.
Vivaldi Antenna Arrays Feed by Frequency-Independent Phase Shifter for High Directivity and Gain Used in Microwave Sensing and Communication Applications. Sensors 2021, 21, 6091. https://doi.org/10.3390/s21186091.
Compact Size and High Gain of CPW-Fed UWB Strawberry Artistic Shaped Printed Monopole Antennas Using FSS Single Layer Reflector," in IEEE Access, vol. 8, pp. 92697-92707, 2020, doi: 10.1109/ACCESS.2020.2995069.
2. The authors mentioned the term 'dual-band antenna.' However, it's not clear from the context how we can determine that it is indeed a dual-band antenna. Additionally, could you please provide information on the frequencies at which the antenna operated?
3. The graph of antenna Return loss is mission?
4- Comparison table with related work is missing.
5- The authors mentioned Figure 15(a). However, there is no such Figure 15(a)!
Moderate editing of English language required.
Author Response
Thank you very much for taking the time to review this manuscript. As you are concerned, there are several problems that need to be addressed. Our research mainly focuses on the research of acquisition control technology, which is not well reflected in our manuscript. Therefore, our manuscript title "Research on Development 3D Ground Penetrating Radar on Dual Band Antennas for Road Underground Dishes" does not reflect our research objectives, So we have revised the title of our manuscript to 'Research on Development 3D Ground Penetrating Radar Acquisition and Control Technology for Road Underground Discuses with Dual Band Antenna Arrays', and accordingly made revisions to the abstract and introduction.
According to your nice suggestions, we have made extensive corrections to our previous draft, the detailed responses to you are listed below.
Comments 1: The introduction needs improvement. The authors should provide more information about the antennas being used for GPR applications. Additionally, please discuss the antenna parameters, such as bandwidth and gain [1,2]. Please refer to these references as they may add value to the introduction.
Vivaldi Antenna Arrays Feed by Frequency-Independent Phase Shifter for High Directivity and Gain Used in Microwave Sensing and Communication Applications. Sensors 2021, 21, 6091. https://doi.org/10.3390/s21186091.
Compact Size and High Gain of CPW-Fed UWB Strawberry Artistic Shaped Printed Monopole Antennas Using FSS Single Layer Reflector," in IEEE Access, vol. 8, pp. 92697-92707, 2020, doi: 10.1109/ACCESS.2020.2995069.
Response 1: Thank you for your suggestion. In order to better represent our research goal of acquisition and control technology for dual band antenna array 3D ground penetrating radar. We have revised the title, abstract, and introduction of the manuscript. More prominently highlight the content of our alliance manuscript. We deeply apologize for not being able to clearly demonstrate our research objectives. We hope to receive your forgiveness.
Comments 2: The authors mentioned the term 'dual-band antenna.' However, it's not clear from the context how we can determine that it is indeed a dual-band antenna. Additionally, could you please provide information on the frequencies at which the antenna operated?
Response 2: Thank you very much for your question. The dual band we mentioned is actually a butterfly antenna with two frequencies, with the center frequencies of 200MHz and 400MHz, respectively, as individual shielded antennas. We have designed a TDSM acquisition and control method using a dual row equidistant array layout, providing data for the fusion of the two antenna echoes in the future. We have included the specific introduction of its antenna in section 3 of the manuscript (page 3).
Comments 3: The graph of antenna Return loss is mission?
Response 2: Thank you very much for the questions and suggestions. We have added the echo loss diagram of a single shielded bow tie antenna in the manuscript, as shown in Figure 3 (page 5).
Comments 4: Comparison table with related work is also missing.
Response 4: Thank you for your request and suggestion. We have added Comparison table with related work in the manuscript, as shown in Table 4 (page 13).
Comments 5: The authors mentioned Figure 15(a). However, there is no such Figure 15(a)!
Response 5: Thank you for your request and suggestion. This is our mistake, and we have made corresponding modifications to your proposal. Please refer to page 13 of the manuscript for details. We deeply apologize for any inconvenience caused to your reading.
We tried our best to improve the manuscript and made some changes marked up using the “Track Changes” function in revised manuscript. After revising, the quality of this manuscript is highly improved. And the expression of the article is clearer than the original manuscript. If you think there is any need for further modification, we will actively cooperate. We hope you will find our revised manuscript acceptable for publication.